# Identification and Comparative Analysis of Long Non-Coding RNA in the Skeletal Muscle of Two Dezhou Donkey Strains

**DOI:** 10.3390/genes11050508

**Published:** 2020-05-04

**Authors:** Tianpei Shi, Wenping Hu, Haobin Hou, Zhida Zhao, Mingyu Shang, Li Zhang

**Affiliations:** Institute of Animal Sciences, Chinese Academy of Agricultural Sciences, Beijing 100193, China; susannashi@163.com (T.S.); huwenping@caas.cn (W.H.); houhaobean@163.com (H.H.); zzzdddddd@163.com (Z.Z.); shangmingyu263@sina.com (M.S.)

**Keywords:** lncRNA, Dezhou donkey, skeletal muscle, TGF-β signaling pathway

## Abstract

Long non-coding RNA (lncRNA) has been extensively studied in many livestock. However, compared with other livestock breeds, there is less research regarding donkey lncRNA function. It has been reported that lncRNA plays an important role in the timing control of development, aging, and death of livestock. Therefore, the study of donkey skeletal muscle lncRNA is of great significance for exploring donkey meat production performance. In this study, RNA-Seq was used to perform high-throughput sequencing of skeletal muscle (longissimus dorsi and gluteus) of two Dezhou donkey strains (SanFen and WuTou). The differentially expressed lncRNAs were screened between different strains and tissues. Then candidate genes for conjoint analysis were screened based on Gene Ontology (GO) and Kyoto Encyclopedia of Genes and Genomes (KEGG) analysis. Finally, the accuracy of the sequencing data was verified by real-time quantitative polymerase chain reaction (RT-qPCR). Herein, we identified 3869 novel lncRNAs and 73 differentially expressed lncRNAs. Through the comparison between groups, the specific expression of lncRNAs were found in different strains and muscle tissues. Importantly, we constructed the lncRNA-miRNA-mRNA interaction network and found three important candidate lncRNAs (MSTRG.9787.1, MSTRG.3144.1, and MSTRG.9886.1) and four candidate genes (*ACTN1*, *CDON*, *FMOD*, and *BMPR1B*). Analysis of the KEGG pathway indicates that the TGF-β signaling pathway plays a pivotal role in the growth and development of skeletal muscle in Dezhou donkey strains.

## 1. Introduction

Long non-coding RNA (lncRNA) distributes widely in mammals, and its expression is both spatiotemporally and tissue specific. LncRNA plays an important role in many processes, such as epigenetic regulation, transcriptional regulation, and post-transcriptional regulation [1]. In the past, it was thought that lncRNA could not translate into protein because of the lack of typical open reading frames (ORFs) [2]. With the development of high-throughput sequencing technology, there is increasing evidence that lncRNA plays various roles in cells and organization [3], like the regulation of mRNA and protein stability [4,5], X-chromosome inactivation [6,7], regulation of miRNA activities [8], regulation of epigenetic modifications [9], and embryonic development [10]. These studies have broadened the traditional understanding of the genetic coding potential of genomes and increased awareness of the diversity of types and functions of lncRNA. Plenty of lncRNAs associated with skeletal muscle development in multiple species has been identified by RNA-seq, such as human (*Homo sapiens*), mouse (*Mus musculus*), pig (*Sus scofa*), cow (*Bos taurus*), goat (*Capra hircus*), and horse (*Equus caballus*). A total of 12,103 pig lncRNAs and 8250 cow lncRNAs are included in the Domestic-Animal Long Noncoding RNAdatabase (ALDB) [11]. The horse lncRNA database has 20,800 transcripts that demonstrate characteristics unique to lncRNAs, including low expression, low exon diversity, and low level of sequence conservation [12]. However, there are few studies on the function of donkey non-coding RNA (ncRNA).

Donkey is a kind of important herbivorous livestock across the world, and has witnessed the changes of history. The modern donkey industry is primarily focused on the utilization of meat, skin, and milk in China. The Dezhou donkey is a typical breed in China which is well known for its tall body, muscular body, well-proportioned structure, good performance, and pure coat color, etc. Donkey foals grow fast, as male and female donkeys can reach 90% and 85% of their adult height and body length when they are 12 months old, and 100% and 95.7% of their annual age when they are 24 months. It has been confirmed that every 100 g of donkey meat contains 18.6 g of protein, 0.7 g of fat, 10 mg of calcium, 144 mg of phosphorus, 136 mg of iron, and 3.347 MJ of calories. Compared with other livestock meats, donkey meat shows “three high and three low” characteristics [13]: high protein, high essential amino acids, high essential fatty acids, and low in fat, low in cholesterol, and low in calories [14]. The meat yield is perfect, and the slaughter rate can be over 50%. Therefore, the development and breeding potential of Dezhou donkey is worth studying. The Dezhou donkeys are divided into two strains according to their obvious and unique appearance features and physical characteristics (China national commission of animal genetics resources, 2011), namely the SanFen donkey (SF) and WuTou donkey (WT). The two strains are both covered with black hair, but the SanFen donkey has white hair on the eyes, around the nose, and under the belly. By analyzing the height, body length, tube circumference, and bust data of SanFen and WuTou from pup donkey to full-grown donkey, we found that the body shape of full-grown SanFen is smaller than that of WuTou, the growth rate of young WuTou is significantly faster than that of SanFen. In slaughter traits, the backfat thickness of WuTou is thicker than that of SanFen, and the net meat rate of WuTou is slightly higher than that of SanFen. The average birth body weight, body height, and body length of WuTou donkey are 26.5 kg, 86 cm, and 59 cm, while those of SanFen donkey are 23.9 kg, 81 cm, and 55 cm. Therefore, we performed high-throughput sequencing of skeletal muscle to establish a skeletal muscle expression profile so that we can explore and try to explain the reason for the differences at the transcriptome level. Two skeletal muscle longissimus dorsi (B) and gluteus (T) were taken as sequencing samples in two strains, and four groups (SB vs. ST, WB vs. WT, WB vs. SB, and WT vs. ST) were compared. Moreover, differentially expressed lncRNAs were selected and annotated through threshold value and bioinformatics analysis. Finally, function prediction and target gene prediction were played in order to mine candidate lncRNAs and hub genes related to muscle development. What we have done helps to accelerate the identification of lncRNAs related to important meat production of Dezhou donkey and analysis of interaction regulation mechanisms in the growth of skeletal muscle in Dezhou donkey. This research will expand the donkey transcript information database and provide a data basis for further exploration of the relationship between donkey genes and transcriptome regulation.

## 2. Results

### 2.1. Overview of RNA Sequencing

We constructed 12 cDNA libraries (SB_1, SB_2, SB_3, ST_1, ST_2, ST_3, WB_1, WB_2, WB_3, WT_1, WT_2, WT_3) with donkey B and T muscle samples from two donkey strains. A total of 24,683 genes and 49,265 transcripts were annotated. More than 88,000,000 raw data were detected in each sample, and 81,142,113 average clean reads per sample were obtained after quality control. The clean data obtained by each sequencing library accounted for about 97% of the raw data, and the reads of Q20 and Q30 exceeded 97%. Approximately 94% of total clean reads were mapped to the donkey reference genome (Table 1), which indicates that the sequencing data are good and can be used for subsequent data analysis. The homology analysis of donkey skeletal muscle lncRNAs with human and mouse showed that donkey and mouse had higher homology. The percentage of identity with human was 91.12% (77.32–100.00%), and the percentage of identity with mouse was 95.34% (77.40–100.00%).

### 2.2. The Characteristics Comparison of Transcripts

In this study, we identified novel lncRNAs using CNCI(Version 2, GitHub, San Francisco, CA, USA) and CPC software (Version 2, CBI, USA), and the intersections of these predictions revealed 3869 novel lncRNA transcripts, including 1569 (40.55%) intergenic lncRNAs (class code “u”), 131 (3.39%) anti-sense lncRNAs (class code “x”), 1946 (50.30%) intronic lncRNAs (class code “i”), 53 (1.37%) generic exonic overlap lncRNAs with reference transcripts (class code “o”), and 170 (4.39%) potentially novel isoform lncRNAs (class code “j”). The proportion of lncRNA class codes in each sample is shown in Appendix A. The CNCI and CPC software score for each sample are shown in Figure 1. There were differences in the degree of differential expression between “u”, “x”, “i”, and “o”, and the expression level of “u” was highest in skeletal muscle (Figure 2). The size of Dezhou donkey lncRNAs ranged from 203 to 56,098 nucleotides, with an average size of approximately 1752 bp. The average exon number of these lncRNAs was 1.3 (82.5% of total lncRNAs have one exon), far less than the average of 12.3 exons for mRNAs (Figure 3a). The expression level and a number of mRNAs and lncRNAs are shown in Figure 3b. ORFs length in identified lncRNAs were shorter (mean 89.69 amino acids) (Figure 3c) than in protein-coding genes (mean 598.71 amino acids) (Figure 3d). In addition, we found that 59.4% of total lncRNAs target genes were found (protein-coding genes located in 100 kb around the lncRNAs).

### 2.3. Identification of Differentially Expressed lncRNAs

Expression level of lncRNA transcripts were estimated by FPKM. According to a comparison between the two strains, 35 lncRNAs were differentially expressed in the longissimus dorsi muscle and 38 lncRNAs were differentially expressed in the gluteus. There were 29 differentially expressed lncRNAs between the two tissues of the WuTou donkey, and 17 differentially expressed lncRNAs between the two tissues of the SanFen donkey (Figure 4a). To further analyze the interactions of differentially expressed lncRNAs in the two strains, Venn maps were constructed using all expressed and differentially expressed lncRNAs from skeletal muscle samples for WB, SB, WT, and ST, respectively (Figure 4b,c). This result revealed 2,452 lncRNAs (72.8%) expressed across all sample groups and that the differential expression of lncRNAs possessed characteristics of strain-specific and muscle-specific. The top 10 most significantly expressed lncRNAs in the WuTou and SanFen donkeys are shown in Table 2 and Table 3, respectively. MSTRG.4504.1, MSTRG.9671.3, and MSTRG.9787.5 were expressed in the longissimus dorsi muscle of WuTou donkeys but were almost completely absent from SanFen donkeys. However, MSTRG.9787.2 and MSTRG.4872.1 were expressed in the longissimus dorsi muscle of SanFen donkey but were almost completely absent from WuTou donkeys. By identifying the tissue-specific differentially expressed lncRNAs between the two strains, 6, 7, 7, and 13 unique differentially expressed lncRNAs were found in the WB, WT, SB, and ST groups, respectively. In the two strains, 13 and 6 specific differentially expressed lncRNAs were found in WuTou and SanFen donkey, respectively. Detailed lncRNAs information is shown in Table 4.

### 2.4. Validation of RNA-Seq Data

The expression level of several lncRNAs were detected by real-time quantitative polymerase chain reaction (RT-qPCR) analysis in order to validate the accuracy of RNA sequencing data. As shown in Figure 5, the results of RT-qPCR were consistent with the sequencing data.

### 2.5. Co-Expression of lncRNAs and mRNAs

The target genes for cis-regulation by lncRNAs were predicted using the location relationship. By screening 100 kb upstream and downstream sites of lncRNAs, candidate lncRNAs were identified. Enrichment or connectivity was due to the position frequency matrix as described in a previous study. Results contained only genes with tightly correlated expression profiles (Pearson’s r ≥ 0.95). Based on the differentially expressed lncRNAs, we performed a target prediction by lncRNA Targets tool. Finally, 8 pairs of lncRNA-mRNA were identified in WB vs. WT group, 14 pairs in SB vs. ST group, 9 pairs in WB vs. SB group, and 17 pairs in WT vs. ST group, respectively. Among them, only MSTRG.14368.3 was differentially expressed lncRNA, and the host gene is *PPFIA1*.

### 2.6. lncRNA–miRNA–mRNA ceRNA Network Construction

Competing endogenous RNA, also known as endogenous competitive binding RNA, is considered a kind function of new regulatory RNA, including lncRNA, circRNA, pseudogene and so on. This RNA regulatory element competes for binding to miRNA, and the target genes regulated by miRNA are consequently upregulated. Based on the differentially expressed lncRNAs data of the group WB vs. SB and group WT vs. ST, miRNA–mRNA and miRNA–lncRNA pairs were targeted to construct the lncRNA–miRNA–mRNA ceRNA network using an in-house Perl script. As shown in Figure 6 and Figure 7, miRNAs shared 208 edges with 89 unique genes and seven unique lncRNAs in the ceRNA network. The ceRNA network was visualized by Cytoscape software (Version 3.8.0, National Resource for Network Biology, USA). Genes within the networks were further processed by gene function (GO and KEGG) pathway analysis. According to the topological characteristics and degree, *ACTN1*, *FMOD*, *NEDD4L*, and *BMPR1B* were screened as potential candidate genes while MSTRG.9787.1 and MSTRG.3144.1 were screened as potential candidate lncRNAs.

### 2.7. Gene Function and Pathway Analysis

To explore the function of the ceRNA network, hub genes within the network were further analyzed with GO and KEGG databases. Based on BiNGO enrichment analysis [15], all GO functions were found to be significant components relating genes in the network involved in processes such as the regulation of transcription from the RNA polymerase II promoter, protein phosphorylation, cytoplasm, nucleus, protein binding, ATP binding, and other biological processes, cellular components, and molecular functions (Figure 7). Functional pathway analysis demonstrated the ceRNA network potentially modulates multiple signaling pathways. 

Pathways related to muscle growth and development, such as the Wnt signaling pathway, TGF-β signaling pathway, and the glucagon signaling pathway, were significantly associated with glycogen decomposition and gluconeogenesis (Figure 8). The TGF-β signaling pathway significantly enriched within ceRNA-regulated genes (Figure 9). Within the pathway, 12 unique genes were enriched. These genes, including *BRD3*, *CDC42BPA*, *CHRD*, *IRF2BPL*, *MAPK4*, *THBS1*, *BMPR1B*, *MAPK6*, *RBL1*, *ROCK1*, *SMAD9*, and *TGIF2*, regulated essential signaling molecules for the TGF-β signaling pathway.

## 3. Discussion

Many studies have described the expression level, annotated functions and expression differences related to cells and tissues in various species. However, few studies have reported the interaction between lncRNAs and other ncRNAs in donkey, except some in horse. In this study, a lncRNA profile using whole transcriptome sequencing was established and the differences of lncRNAs in four comparison groups (WB vs. SB, WT vs. ST, WB vs. WT, and SB vs. ST) were identified. The ultimate result is that 3869 novel lncRNAs were identified, and the subtype lncRNAs and the characteristics between lncRNAs and mRNAs were clarified [16]. Which confirmed our results such as structural features and expression level are consistent with some previous studies in other animals [17,18]. For example, lncRNAs contain shorter putative ORFs and lower expression level than ttat of mRNAs [19,20]. We also found that 90.85% lncRNAs were intergenic (40.55%) or intronic (50.30%), which is in accordance with the previous findings that intergenic and intronic regions are the major sources of lncRNAs. Through comparison between groups, we found that the expression of lncRNAs in skeletal muscle of two Dezhou donkey strains (SF and WT) was quite different, while the expression of lncRNAs in different tissues (B and T muscle) of the one strain was only slightly different. 

To further research the function of the novel lncRNAs and differentially expressed lncRNAs, we performed target gene predictions based on the possible mode of action of lncRNAs, such as cis and trans-regulatory, lncRNAs-bound nucleic acid sequences, which provided an effective method to infer their potential function. However, due to the assembly of imperfect donkey genomes, the annotated lncRNAs and target genes are limited. GO and KEGG analysis were used for identifying which biological process major lncRNAs participated in. The analysis of GO categories presents that most differentially expressed lncRNAs involved in many processes. In addition, we found the hub lncRNAs, miRNAs, and candidate genes that differentially expressed among comparison groups involved in glucagon signaling, fatty acid biosynthesis, TGF-β signaling pathways were found. Their pathways are important to study the regulation in the development of skeletal muscle, including glucose metabolism, lipid metabolism, and skeletal muscle development.

In the process of co-expression network construction and function prediction, several differentially expressed genes in the longissimus dorsi muscle caught our attention in longissimus dorsi muscle. GO analysis showed that the *ACTN1* gene was related to skeletal muscle development, *NR4A1* and *EGR1* genes were associated with skeletal muscle cell differentiation, and the *EZH2* gene was involved in skeletal muscle satellite cell maintenance in skeletal muscle regeneration. Therefore, lncRNA (*RCOR1*: MSTRG.9886.1)-miRNA (efu-mir-9341-p3_1ss14GC)-mRNA (*NR4A1*) may play a key role in the ceRNA network. In gluteus muscle, the *CDON* gene was related to skeletal muscle satellite cell differentiation, the *ACTC1* gene was related to skeletal muscle filament assembly, the *BCL9* gene was involved in skeletal muscle cell differentiation, and *FAM65B* was involved in skeletal muscle fiber development. The data were used to perform KEGG enrichment analysis, and the TGF-β signaling pathway is significantly enriched. Studies have shown that manipulation of the TGF-β signaling pathway can increase muscle growth in mice [21]. In the lncRNA-miRNA-mRNA network, we screened candidate genes *ACTN1*, *FMOD*, *NEDD4L*, and *BMPR1B*, of which *FMOD* and *BMPR1B* genes belong to the TGF-β signaling pathway. *FMOD* can participate in the regulation of fibrils in the connective tissue and maintain *MSTN* transcriptional activity in skeletal muscle tissue. In addition, the expression of *MYOG* and *MYL2* in the tissue are regulated by the *FMOD* gene [22,23]. *BMPR1B* protein is a signal transduction protein. The function of *BMPR1B* is involved in mammalian ovarian follicle development, animal embryonic development, bone tissue formation, cancer cell growth, and brain tissue recovery [24]. At present, the *BMPR1B* gene has been regarded as an important candidate gene for controlling the inheritance of the reproductive performance of sheep [25]. The host gene of MSTRG.9787.1 is *ARF6*, and *ARF6* has been associated with proliferation in many cell lines. It is reported that *ARF6* can control *ROS* production to mediate angiotensin II-Induced vascular smooth muscle cell proliferation [26]. MSTRG.3144.1 was encoded by *ZBTB20* which belongs to zinc finger transcription factors. *ZBTB20* can act as a transcriptional repressor of *FOXO1* to promote cell proliferation and tumor growth. *ZBTB20* null mice exhibit severe postnatal growth retardation, metabolic dysfunction, and lethality, which suggested that *ZBTB20* plays an essential role in animal embryonic development [25]. Therefore, we selected lncRNAs (MSTRG.9787.1MSTRG.3144.1, and MSTRG.9886.1) and genes (*ACTN1*, *CDON*, *FMOD*, and *BMPR1B*) as important candidate genes through functionl analysis. In addition, our KEGG analysis identified pathways related to insulin or glucagon, which may indirectly affect the growth and development of skeletal muscle.

## 4. Materials and Methods

### 4.1. Sample Collection and Preparation

Longissimus dorsi (B) and gluteus (T) muscle samples were taken from 3 male SanFen donkeys and 3 male WuTou donkeys which were healthy and had no genetic relationship with each other. The donkeys were raised in the same feedlot and fed with the same total mixed ration (TMR). The height, body length, tube circumference and bust were measured and humanely sacrificed in an accredited commercial slaughterhouse when they reached 24 months old to collect skeletal muscle samples. All samples were immediately frozen in liquid nitrogen and stored at −80 °C before the experiment. All animal protocols were approved by the National Donkey Breeding Center of Dong’e, Shandong. All animal work here was approved by the Animal Welfare & Ethics Committee of Institute of Animal Sciences, Chinese Academy of Agricultural Sciences (NO. IAS2019-22).

### 4.2. RNA Isolation and Sequencing

Total RNA was extracted using TRIzol reagent (Invitrogen, CA, USA) following the manufacturer’s protocol. Total RNA quantity and purity were assessed using a Bioanalyser 2100 and an RNA 6000 Nano LabChip Kit (Agilent, CA, USA) with RIN number >7.0. According to the manufacturer’s instruction, approximately 10 µg total RNA was prepared to construct the library. Ribosomal RNA was removed according to the Epicentre Ribo-Zero Gold Kit (Illumina, San Diego, CA, USA). Following purification, the poly (A)− or poly (A)+ RNA fractions were fragmented into small pieces using divalent cations under elevated temperature. Then, cleaved RNA fragments were reverse-transcribed to create the final cDNA library in accordance with the protocol for the mRNA-Seq sample preparation kit (Illumina, San Diego, CA, USA), the average insert size for the paired-end libraries was 300 bp (±50 bp). Finally, we performed paired-end sequencing on an Illumina Hiseq 4000 with (lc-bio, China) the vendor’s recommended protocol.

### 4.3. Transcripts Assembly

First, Cutadapt [21] was used to remove adaptor contamination, low-quality bases, and undetermined bases. Then, sequencing quality was verified using FastQC. We used Bowtie2 [21] and Tophat2 [27] to map reads to the genome of donkey, and assembled the reads for each sample using StringTie [28]. Next, all transcriptomes from donkey samples were merged to reconstruct a comprehensive transcriptome using Perl scripts. After the final transcriptome was generated, StringTie and Ballgown [7] were used to estimate expression level.

### 4.4. LncRNA Identification

First, transcripts that overlapped with known mRNAs and transcripts shorter than 200 bp were discarded. Next, we utilized CPC [29], CNCI [30] and Pfam [31] to predict transcripts with coding potential. All transcripts with CPC score <−1 and CNCI score <0 were removed. The remaining transcripts with class codes (i: A transfrag falling entirely within a reference intron; j: Potentially novel isoform (fragment): at least one splice junction is shared with a reference transcript; o: Generic exonic overlap with a reference transcript; u: unknown, intergenic transcript; x: Exonic overlap with reference on the opposite strand) were considered lncRNAs.

### 4.5. Differential Expression Analysis of mRNAs and lncRNAs

StringTie was used to perform expression level analysis for mRNAs and lncRNAs by calculating fragments per kilobase million (FPKM). Differentially expressed mRNAs and lncRNAs were selected using |log2 (fold change)| ≥ 1 and with statistical significance (*p*-value ≤ 0.05) [32].

### 4.6. Target Gene Prediction and Function Analysis of lncRNAs

To explore the function of lncRNAs, we predicted their cis-target genes, as lncRNAs may play a cis role, acting on neighboring target genes. In this study, coding genes 100,000 bp upstream and downstream were selected by Perl script. Then, we performed a function analysis of the target genes for lncRNAs using the scripts in the house. Significance is expressed as a *p*-value ≤ 0.05.

### 4.7. Prediction of miRNAs Targets of lncRNAs and Target Genes

The lncRNAs and 3′ UTR sequences of its target genes were predicted as miRNA targets using TargetScanand Miranda [33]. These lncRNAs were regarded as competing for endogenous RNAs (ceRNA) that competitively bind miRNA and indirectly affect mRNA (miRNA target gene) expression. The ceRNA-regulating cascades (lncRNA-miRNA-mRNA) were built using home Perl scripts.

### 4.8. Validation of RNA-Seq Data

The expression of several lncRNAs was detected by RT-qPCR analysis in order to validate the accuracy of RNA sequencing data. The primers used for the RT-qPCR were listed below (Appendix A). RT-qPCR was performed using SYBP Green (TaKaRa Biotech, Dalian, China) according to the manufacturer’s protocol.

## 5. Conclusions

This study is the first time to identification and comparative analysis of lncRNA of skeletal muscle in different donkey strains. Firstly, we identified 3869 novel lncRNAs in skeletal muscle tissues (B and T) from two strains of Dezhou donkey which have never been researched. In addition, 35 and 38 differentially expressed lncRNAs were identified in the longissimus dorsi and gluteus muscle of Dezhou donkey, respectively. Lastly, several remarkable lncRNAs (MSTRG.9787.1, MSTRG.3144.1, and MSTRG.9886.1) and genes (*ACTN1*, *CDON*, *FMOD*, and *BMPR1B*) of the TGF-β signaling pathway were found using function prediction analysis and co-expression network was constructed, which play an important role in the growth of donkey skeletal muscle. To sum up, all the results do provide good ideas for further revealing donkey skeletal muscle developmental mechanism, improving the donkey meat production performance, and carrying marker assisted selection in donkey breeding. Our studies also encourage more donkey research for the future development of donkey industry.

## Figures and Tables

**Figure 1 genes-11-00508-f001:**
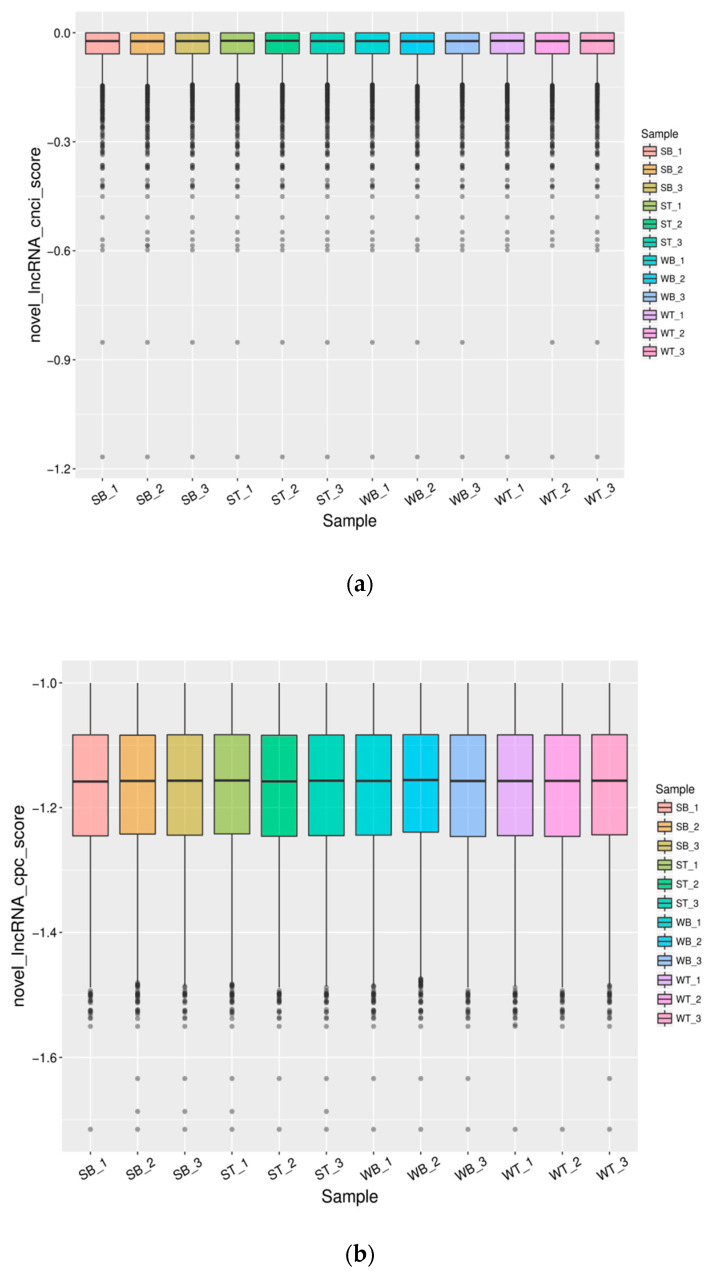
Boxplot of the novel lncRNAs prediction score (**a**): Novel lncRNAs prediction score wih CNCI software (Version 2, GitHub, San Francisco, CA, USA). (**b**): Novel lncRNAs prediction score wih CPC software (Version 2, CBI, USA).

**Figure 2 genes-11-00508-f002:**
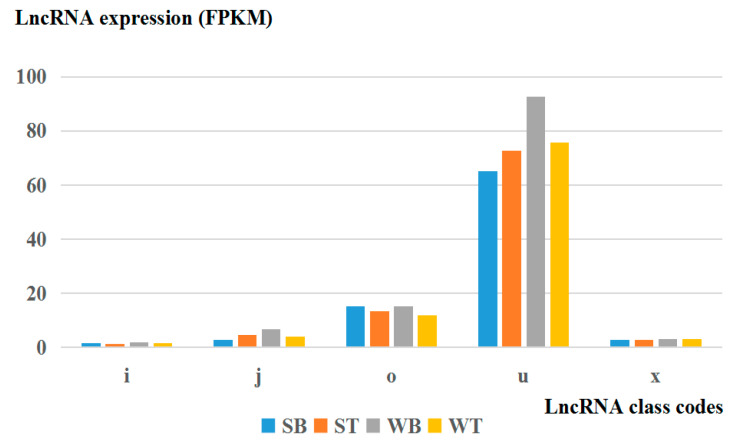
Different class codes of expression level of skeletal muscle lncRNAs. potentially novel isoform (fragment): at least one splice junction is shared with a reference transcript; i: a transfrag falling entirely within a reference intron; o: generic exonic overlap with a reference transcript; u: unknown, intergenic transcript; x: exonic overlap with reference on the opposite strand; WB, WT, SB, and ST represent WuTou donkey longissimus dorsi muscle, WuTou donkey gluteus muscle, SanFen donkey longissimus dorsi muscle and SanFen donkey gluteus muscle, respectively.

**Figure 3 genes-11-00508-f003:**
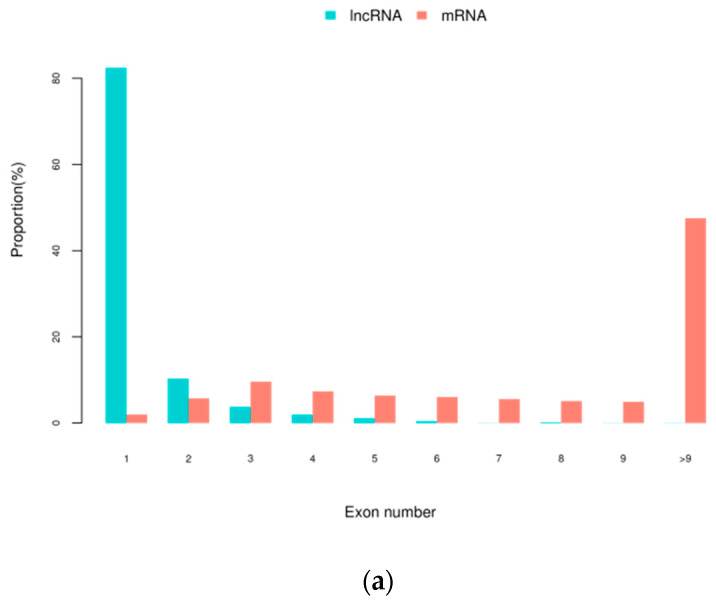
The Characteristics Comparison of Transcripts (**a**): Distribution of exon number for mRNAs and lncRNAs. (**b**): Expression level and a number of mRNAs and lncRNAs. (**c**): Distribution of lncRNA ORFs length. (**d**): Distribution of mRNA ORFs length.

**Figure 4 genes-11-00508-f004:**
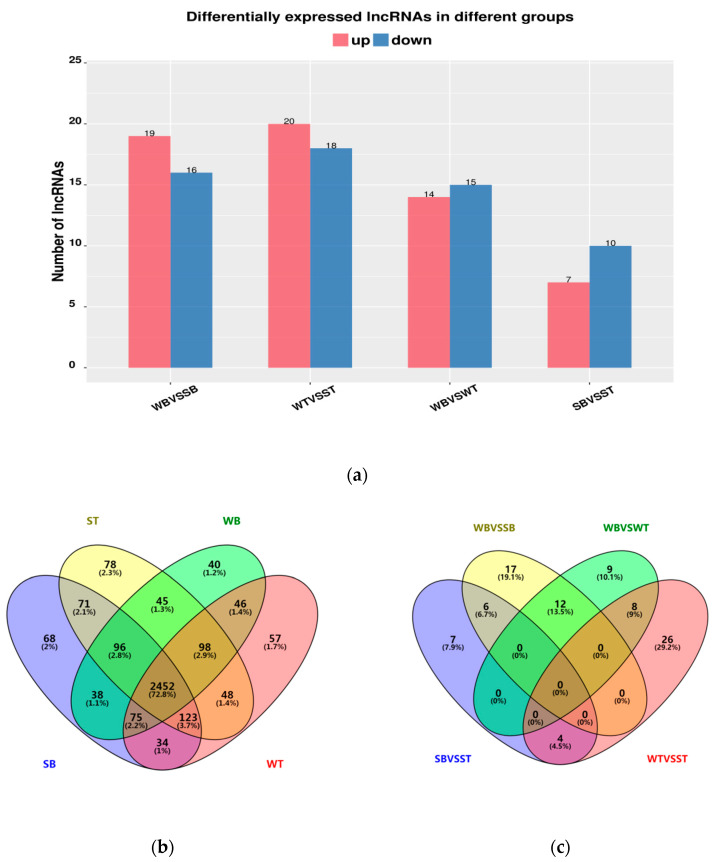
lncRNAs bar chart and venn diagram. (**a**): Differentially expressed lncRNAs in different groups. (**b**): All expressed lncRNAs venn diagram among the four groups. (**c**): Differentially expressed lncRNAs venn diagram among the four groups (WB vs. SB, WT vs. ST, WB vs. WT, SB vs. ST). WB, WT, SB, and ST represent WuTou donkey longissimus dorsi muscle, WuTou donkey gluteus muscle, SanFen donkey longissimus dorsi muscle, and SanFen donkey gluteus muscle, respectively.

**Figure 5 genes-11-00508-f005:**
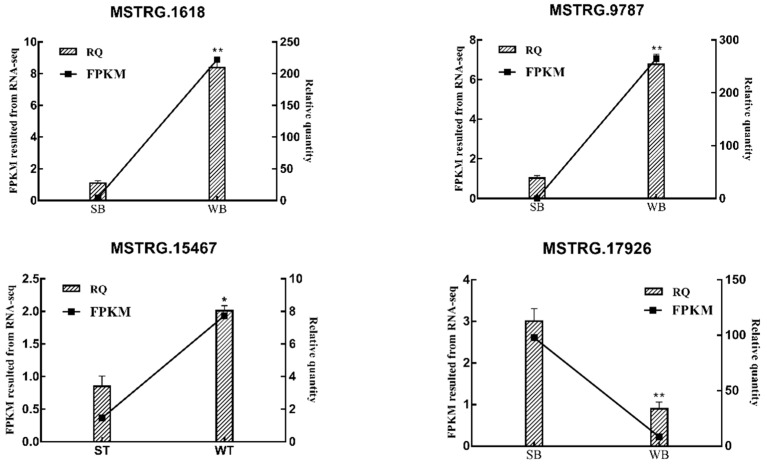
RT-qPCR results of lncRNAs with significant differences in WB vs. SB and WT vs. ST comparison group. The figure shows the quantitative results of RNA-seq and RT-qPCR; the line graph shows the FPKM value of the sequencing and the histogram shows the relative quantitative results; the significance test: * *p* ≤ 0.05, ** *p* ≤ 0.01. WB, WT, SB, and ST represent WuTou donkey longissimus dorsi muscle, WuTou donkey gluteus muscle, SanFen donkey longissimus dorsi muscle and SanFen donkey gluteus muscle, respectively.

**Figure 6 genes-11-00508-f006:**
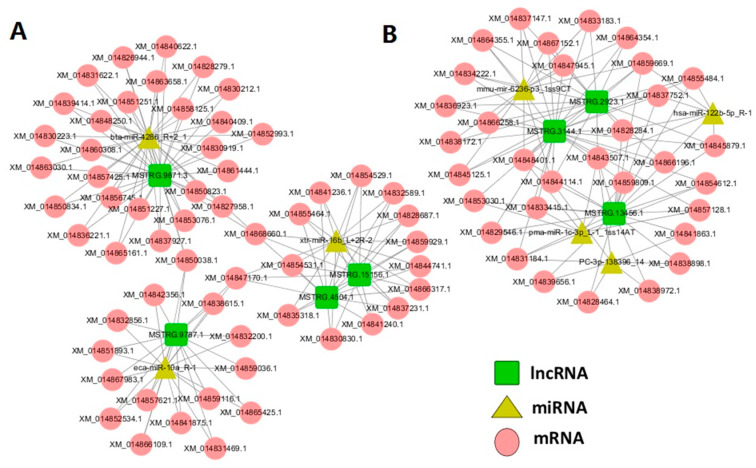
The ceRNA network. Green squares represent lncRNAs, yellow triangles represent miRNAs, and red circles represent mRNAs. (**A**): WB vs. SB. (**B**): WT vs. ST. WB, WT, SB, and ST represent WuTou donkey longissimus dorsi muscle, WuTou donkey gluteus muscle, SanFen donkey longissimus dorsi muscle and SanFen donkey gluteus muscle, respectively.

**Figure 7 genes-11-00508-f007:**
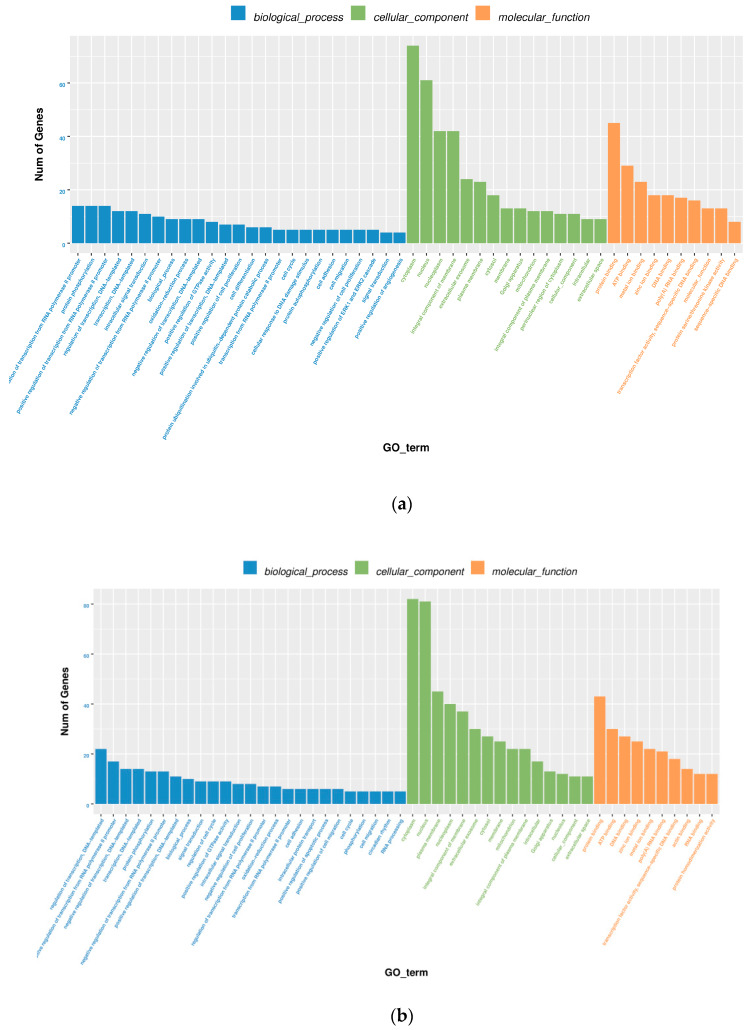
GO enrichment analysis of the ceRNA network. (**a**): WB vs. SB. (**b**): WT vs. ST. WB, WT, SB, and ST represent WuTou donkey longissimus dorsi muscle, WuTou donkey gluteus muscle, SanFen donkey longissimus dorsi muscle, and SanFen donkey gluteus muscle, respectively.

**Figure 8 genes-11-00508-f008:**
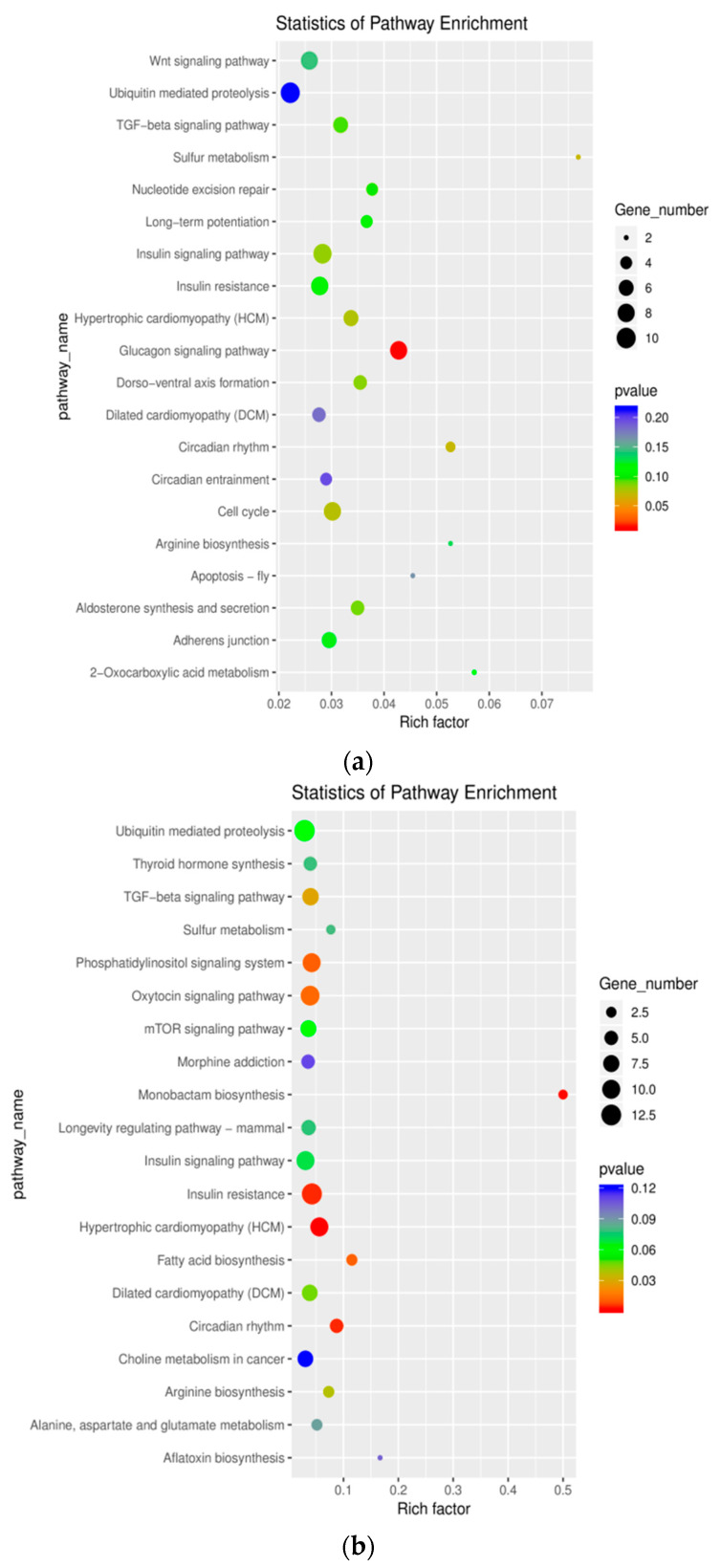
Significantly enriched KEGG terms of the ceRNA network. (**a**): WB vs. SB. (**b**): WT vs. ST. Y-axis: pathway name; X-axis: enrichment factor; Bubble size reflects the number of differential expressions enriched to a certain pathway/all quantities in the background; Color reflects the significance of differential expression in a certain process. WB, WT, SB, and ST represent WuTou donkey longissimus dorsi muscle, WuTou donkey gluteus muscle, SanFen donkey longissimus dorsi muscle, and SanFen donkey gluteus muscle, respectively.

**Figure 9 genes-11-00508-f009:**
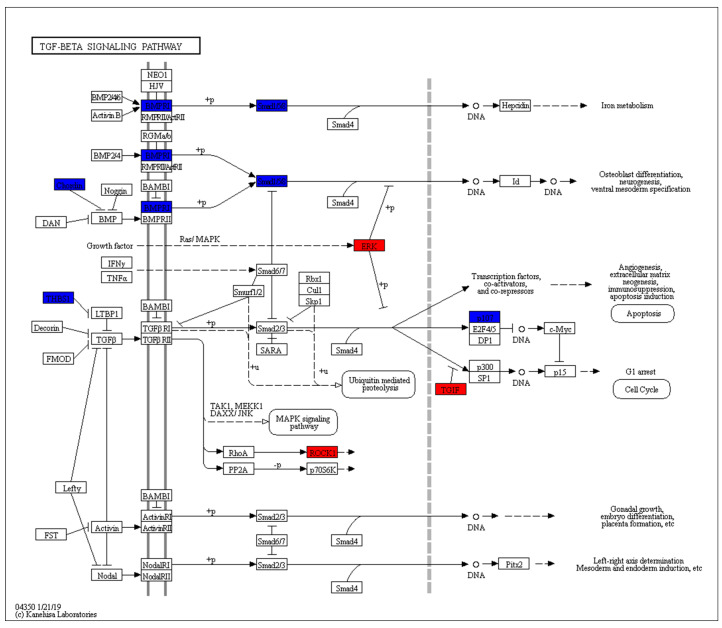
TGF-β signaling pathway. Red indicates up-regulated genes and blue indicates down-regulated genes.

**Table 1 genes-11-00508-t001:** Quality statistics of different sample sequencing data.

Sample	Raw Data	Valid Data	Valid Ratio (%)	Q20 (%)	Q30 (%)	Mapped Ratio (%)
SB_1	92,359,068	89,451,180	96.85	99.61	97.90	94.46
SB_2	89,593,740	86,962,880	97.06	99.50	97.66	94.44
SB_3	92,880,616	90,281,074	97.20	99.72	98.21	93.26
ST_1	91,765,874	89,052,976	97.04	99.63	97.96	94.78
ST_2	91,353,800	89,377,270	97.84	99.56	97.78	94.53
ST_3	88,677,322	86,657,418	97.72	99.81	98.35	88.73
WB_1	91,252,356	89,372,624	97.94	99.57	97.83	94.08
WB_2	88,681,618	86,415,030	97.44	99.62	97.88	93.32
WB_3	91,319,442	89,166,274	97.64	99.67	97.95	95.93
WT_1	88,930,000	86,281,442	97.02	99.58	97.83	94.05
WT_2	91,398,066	89,005,890	97.38	99.71	98.01	91.94
WT_3	92,230,622	89,494,092	97.03	99.64	97.90	95.87

Note: WB, WT, SB, and ST represent WuTou donkey longissimus dorsi muscle, WuTou donkey gluteus muscle, SanFen donkey longissimus dorsi muscle, and SanFen donkey gluteus muscle, respectively.

**Table 2 genes-11-00508-t002:** The top 10 most significantly expressed lncRNAs of longissimus dorsi muscle.

lncRNA ID	Class Code	WuTou (FPKM)	SanFen (FPKM)	*p*-Value	Regulation
MSTRG.4504.1	i	0.87	0.00	0.00	up
MSTRG.9671.3	u	2678.78	0.00	0.00	up
MSTRG.9787.5	u	7173.04	0.00	0.00	up
MSTRG.9787.2	u	0.00	5775.89	0.00	down
MSTRG.4872.1	i	0.00	0.90	0.00	down
MSTRG.9787.13	u	20,614.20	156.14	0.00	up
MSTRG.9787.1	o	264.72	0.06	0.00	up
MSTRG.15156.1	i	0.00	0.69	0.00	down
MSTRG.17523.1	u	0.00	1.01	0.00	down
MSTRG.169.2	j	1.81	0.00	0.00	up

**Table 3 genes-11-00508-t003:** The top 10 most significantly expressed lncRNAs of gluteus muscle.

lncRNA ID	Class Code	WuTou (FPKM)	SanFen (FPKM)	*p*-Value	Regulation
MSTRG.9787.3	u	0.00	5846.54	0.00	down
MSTRG.3144.1	i	0.00	1.96	0.00	down
MSTRG.13456.1	u	0.00	1.06	0.00	down
MSTRG.2923.1	i	1.01	0.00	0.00	up
MSTRG.12585.15	x	3.35	0.00	0.00	up
MSTRG.1618.2	i	0.00	35.11	0.00	down
MSTRG.14368.3	i	0.26	0.00	0.00	up
MSTRG.12391.1	i	0.00	0.80	0.00	down
MSTRG.10272.1	i	0.00	1.46	0.00	down
MSTRG.11396.1	u	0.00	0.32	0.00	down

**Table 4 genes-11-00508-t004:** The tissue-specific differentially expressed lncRNAs in different strains and tissues.

Group	The Tissue-Specific Differentially Expressed lncRNAs			
	WB	WT	SB	ST
WB vs. WT	MSTRG.9671.3, MSTRG.9787.5, MSTRG.9787.1, MSTRG.13456.1, MSTRG.838.1, MSTRG.11396.1, MSTRG.12391.1, MSTRG.8147.1, MSTRG.15510.1	MSTRG.9787.2, MSTRG.4013.5, MSTRG.837.4, MSTRG.17523.1		
SB vs. ST			MSTRG.2923.1, MSTRG.12393.1, MSTRG.2457.1	MSTRG.4504.1, MSTRG.10272.1, MSTRG. 16296.2
WB vs. SB	MSTRG.4504.1, MSTRG.9671.3, MSTRG.9787.5, MSTRG.8147.1, MSTRG.16479.3, MSTRG.15510.1		MSTRG.9787.2, MSTRG.4872.1, MSTRG.15156.1, MSTRG.17523.1, MSTRG.19344.4, MSTRG.9722.1, MSTRG.6042.1	
WT vs. ST		MSTRG.2923.1, MSTRG.14368.3, MSTRG.10853.1, MSTRG.9603.1, MSTRG.3550.1, MSTRG.18276.1, MSTRG.2457.1		MSTRG.9787.3, MSTRG.3144.1, MSTRG.13456.1, MSTRG.1618.2, MSTRG.12391.1, MSTRG.10272.1, MSTRG.11396.1, MSTRG.16828.1, MSTRG.1618.6, MSTRG.9279.1, MSTRG.4133.1, MSTRG.4197.1, MSTRG.18767.1

Note: WB, WT, SB, and ST represent WuTou donkey longissimus dorsi muscle, WuTou donkey gluteus muscle, SanFen donkey longissimus dorsi muscle, and SanFen donkey gluteus muscle, respectively.

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
