# Peer review of "Identification and Comparative Analysis of Long Non-Coding RNA in the Skeletal Muscle of Two Dezhou Donkey Strains"

_genes, 2020, doi:10.3390/genes11050508_

Round 1

Reviewer 1 Report

Generally, the paper deals with the topic of high importance. It brings a lot of new knowledge. However, the presentation of the results must be bettered. It concernes mainly the tables and figures, they must be better arranged.

Some comments:

Tab. 1, describe better the content, and/or add the legend.

The figures are difficult readable and poorly arranged. You should enlarged it and arranged it better. Describe the content in detail, write better the legend. Do not use the abbreviations in the legend.

The list of diferentially expressed lncRNAs in rr. 118-134 is disorganized. Describe shortly the differences, and the list of lncRNAs give only in tables.

In Discussion section, the abbreviations of genes must be in italic.

The last sentence of Conclusion section should be: …which play an important role and may be of great significance in the growth of skeletal muscle in donkey.

Author Response

April 28, 2020

Respected Reviewer,

Thanks to you for the comments on the manuscript. We found that all comments were very helpful and strengthened the manuscript. We took all the  comments into considerations and all amendments were shown in track-changes mode “edited version”(Please see the attachment). Our changes addressing the comments are outlined below. The comments are in italic. In our response, the line numbers and page numbers refer to the revised manuscript unless otherwise stated.

I hope we have adequately addressed your comments and that the manuscript is found to be suitable for publication in your journal.

Point1, describe better the content, and/or add the legend.

Response:Thanks for the reviewer’s suggestion. We have added a note below Table 1 and added the table description on lines 66 in the result section of the revised article (clean format).

 Point2, The figures are difficult readable and poorly arranged. You should enlarged it and arranged it better. Describe the content in detail, write better the legend. Do not use the abbreviations in the legend.

Response:Thanks for the reviewer’s suggestion. We have adjusted all the pictures, enlarged, arranged and described them better. And we explained the abbreviations in all the legends.

 Point3, The list of diferentially expressed lncRNAs in rr. 118-134 is disorganized. Describe shortly the differences, and the list of lncRNAs give only in tables.

Response:Thanks for the reviewer’s suggestion. The reviewer's opinion is very helpful to the modification of this paragraph. We have add a new table (Table 4) to describe the tissue-specific differentially expressed lncRNAs. And we replaced the content of initial maniscript lines 118-134 with "By identifying the tissue-specific differentially expressed lncRNAs between the two strains, 6, 7, 7, 13 unique differentially expressed lncRNAs were found in the WB, WT, SB, ST groups. In the two strains, 13 and 6 specific differentially expressed lncRNAs were found in WuTou and SanFen donkey, respectively. Detailed lncRNAs information is shown in Table 4. " (lines 139-144).

Point4, In Discussion section, the abbreviations of genes must be in italic.

Response:We adjusted the gene fonts in the discussion section and the other part of the manuscript.

Point5, The last sentence of Conclusion section should be: …which play an important role and may be of great significance in the growth of skeletal muscle in donkey.

Response:Thanks for the reviewer’s suggestion. We have changed the last sentence of conclusion section to “Through function prediction analysis and co-expression network construction, we found several remarkable lncRNAs (MSTRG.9787.1, MSTRG.3144.1, and MSTRG.9886.1), genes (ACTN1, CDON, FMOD, and BMPR1B), and the TGF-βsignaling pathway which play important roles and may be of great significance in the growth of skeletal muscle in donkey.” 

Reviewer 2 Report

 The authors performed a standard RNA-seq analysis on total RNA isolated from two different muscles, using two strains of donkey. 3 replications of each muscle from each was used. The purpose of this analysis was to both identify lncRNAs that are present in the donkey, as well as to gain a preliminary view of whether any of these lncRNAs might be differentially expressed between strains, or muscles.

The study as designed is pretty limited is scope. This sampling can only really produce a very preliminary view of any differential expression between samples. The extremely large number of differential expressions is a concern, almost certainly a sign that most of those changes (even if validated over time and many more samples) will be meaningless, considering that the phenotypes of the target animals is not very striking. All of the bioinformatic analysis of target genes, and gene expression pathways are so preliminary that they carry very little value. The authors have RNA-Seq data with which they can perform a first test of the validity of any of the proposed gene networks results in a differential regulation. However it does not seem that expression of the target genes was factored into their analysis, as these predictions are all based on sequences. Also it would be worthwhile to independently interrogate the differential expression of mRNAs between samples to see whether the same ontology pathways are differentially expressed as predicted by their gene network analysis. 

There is value in the study as being direct evidence that lncRNAs can be identified donkey, although I don't consider that there are too many investigators that would consider this a very big advance. The rest of the analysis presented is not very useful, and without experimental evidence its remains speculative at best. 

Although I found the content of manuscript to be of limited interest, the quality of the presentation was good, and everything presented seems scientifically sound. The manuscript was well written, and most every aspect was clearly explained and the visualizations presented were appropriate. 

Author Response

April 28, 2020

Respected Reviewer,

Thanks to you for the comments on the manuscript. We found that all comments were very helpful and strengthened the manuscript. We took all the comments into considerations and all amendments were shown in track-changes mode “edited version”. Our changes addressing the comments are outlined below. The comments are in italic. In our response, the line numbers and page numbers refer to the revised manuscript unless otherwise stated.

I hope we have adequately addressed your comments and that the manuscript is found to be suitable for publication in your journal.

Point: The authors performed a standard RNA-seq analysis on total RNA isolated from two different muscles, using two strains of donkey. 3 replications of each muscle from each was used. The purpose of this analysis was to both identify lncRNAs that are present in the donkey, as well as to gain a preliminary view of whether any of these lncRNAs might be differentially expressed between strains, or muscles.

The study as designed is pretty limited is scope. This sampling can only really produce a very preliminary view of any differential expression between samples. The extremely large number of differential expressions is a concern, almost certainly a sign that most of those changes (even if validated over time and many more samples) will be meaningless, considering that the phenotypes of the target animals is not very striking. All of the bioinformatic analysis of target genes, and gene expression pathways are so preliminary that they carry very little value. The authors have RNA-Seq data with which they can perform a first test of the validity of any of the proposed gene networks results in a differential regulation. However it does not seem that expression of the target genes was factored into their analysis, as these predictions are all based on sequences. Also it would be worthwhile to independently interrogate the differential expression of mRNAs between samples to see whether the same ontology pathways are differentially expressed as predicted by their gene network analysis.

There is value in the study as being direct evidence that lncRNAs can be identified donkey, although I don't consider that there are too many investigators that would consider this a very big advance. The rest of the analysis presented is not very useful, and without experimental evidence its remains speculative at best.

Although I found the content of manuscript to be of limited interest, the quality of the presentation was good, and everything presented seems scientifically sound. The manuscript was well written, and most every aspect was clearly explained and the visualizations presented were appropriate.

Response:Thank you for the valuable suggestions and guidance, the comments are very helpful to the quality of our manuscript. Frankly speaking, our research is a preliminary study for identifying lncRNAs in the skeletal muscle of two Dezhou donkey strains. According to your opinion, we have added the phenotype data of meat production trait in two donkey strains in the introduction section of the manuscript (Line 66-68). There are really some differences between two donkey strains in the Dezhou donkey population, and the purpose of this experiment is just to discover the differences between them by using transcriptome sequencing, enrich the lncRNA database, and initially screen a batch of factors related to skeletal muscle growth, which helps us to verify and reveal the co-expression network analysis.

At present, few studies have focused on donkey muscle development mechanism  and studies on non-coding RNA of donkey are barely noticed. This is the first time to study the Chinese donkey breeds, which means it is a exploratory research. Donkey meat is famous and popular for its special characteristics and being different from other meat, however the meat prodution of donkey is very low. That’s why we are doing such a research. This study provides basic data for donkey genes and transcripts information, which is useful for improving the donkey meat production, donkey breeding, even the industry.

We are going to analyze the mRNA, miRNA and circRNA in donkey muscle next step, and plan to carry out in-depth research and functional verification of genes and lncRNAs as soon as possible. We do hope to have a chance to get your advice again next time.
